# Development of a Hydrologic and Water Allocation Model to Assess Water Availability in the Sabor River Basin (Portugal)

**DOI:** 10.3390/ijerph16132419

**Published:** 2019-07-08

**Authors:** Regina Maria Bessa Santos, Luís Filipe Sanches Fernandes, Rui Manuel Vitor Cortes, Fernando António Leal Pacheco

**Affiliations:** 1Centre for the Research and Technology of Agro-Environment and Biological Sciences, University of Trás-os-Montes and Alto Douro, Ap. 1013, 5001-801 Vila Real, Portugal; 2Chemistry Research Centre, University of Trás-os-Montes and Alto Douro, Ap. 1013, 5001-801 Vila Real, Portugal

**Keywords:** SWAT, MIKE HYDRO Basin, irrigation demand, applied irrigation, irrigation deficit, domestic consumption, resident population

## Abstract

The Sabor River basin is a large basin (3170 km^2^) located in the northeast of Portugal and used mostly for agroforestry. One problem this basin faces is a lack of water during the dry season, when there is a higher demand for water to irrigate crops. To solve this problem, the Portuguese government created a National Irrigation Program to finance new irrigation areas and improve existing ones. Consequently, it is necessary to evaluate the past and future water availability for agricultural and domestic consumption in the basin. This was done through the development of a hydrological and water allocation model. The Soil and Water Assessment Tool (SWAT) was used to model the hydrological processes that took place in the catchment between 1960 and 2008. The MIKE HYDRO Basin was used to simulate water allocation (irrigation and domestic consumption) in a historical view and under two scenarios. The historical view used the time period 1960–2008, and the two scenarios used the same time period but with an increase in the irrigated area. The first scenario simulated the irrigation of the total irrigable area that exists in the basin. The second scenario simulated a 29% increase in the olive grove area and a 24% decrease in the resident population, according to the projection for 2060. The results show that, in the historical view, the average annual water demand deficit was 31% for domestic consumption and 70% for irrigation, which represent 1372 × 10^3^ m^3^ and 94 × 10^6^ m^3^ of water, respectively. In the two scenarios, the water demand deficit increased to 37% for domestic consumption and 77% for irrigation. In the first scenario, the average annual water demand deficit was 183 × 10^6^ m^3^ of water for irrigation. In the second scenario, the average annual water demand deficit was 385 × 10^3^ m^3^ of water for domestic consumption, and 106 × 10^6^ m^3^ of water for irrigating the expanded olive grove area. These results demonstrate that Portuguese farmers can use our model as a decision support tool to determine how much water needs to be stored to meet the present and future water demand.

## 1. Introduction

Portugal is a country with an average annual rainfall of around 700 mm. However, irregular distribution generates water scarcity problems from April to September, particularly in the south and (central and northern) interior of the country. As a result, the spring–summer crops do not have sufficient irrigation to ensure a satisfactory level of production. In this respect, irrigation is a fundamental component of agriculture, without which it is not possible to suitably develop the spring and summer crops.

To solve this problem, the Portuguese government created a National Irrigation Program. The program will make available 534 million euros to create 90,000 hectares of irrigation area with the aim of rehabilitating and modernizing the current irrigation system [1]. The Sabor River basin is included in the national irrigation program, and one of the main objectives of farmers is the expansion of its olive grove area.

The Sabor River basin is located in the Trás-os-Montes region, where the main economic activities are agriculture and livestock farms [2]. This region is dominated by farms that specialize in permanent crops (57%), such as vineyards (24%), fruit trees (8%), olive groves (9%), and combinations of them (16%). The spatial distribution of these crops shows a concentration of viticulture in the Douro Demarcated Region, olive groves in the centre of the region, which is called “warm land”, and a dispersion of farms that specialize in nut, chestnut, and almond trees [3]. According to the agricultural census [4,5,6], the total area of the olive groves in the Trás-os-Montes region increased by 135 km^2^ between 1989 and 2008. This crop is essentially a dryland crop, as only 7% of the area is irrigated according to the agricultural census of 1999 [6].

The general aims of this study are: (i) to determine whether the quantity of water that was produced in the basin between 1960 and 2008 was sufficient to satisfy the requirements of the crops; and (ii) to determine how much water would be needed if the irrigation area was to be increased according to the proposal in the National Irrigation Program. To achieve these aims, we create two scenarios: one to determine how much water is required to irrigate the entire irrigable area that currently exists in the basin, and one based on the expansion of the olive grove area according to a projection for 2060. This study also included the evolution of the resident population between 1960 and 2008, and a demographic projection to 2060.

In order to study the feasibility of water demand, a hydrological and allocation analysis of the basin’s response to agricultural practices and water consumption was carried out. This analysis included a study of the hydrological characteristics of the basin and the development of a hydrological and water allocation model for the basin area. With the advancement of modeling technology, several hydrological models have been developed, ranging from lumped conceptual to physically based distributed models [7]. Watershed models have become a crucial tool for analyzing a wide variety of hitches in water resources, including planning, development, and management. In this study, we propose the use of two separate software packages: Soil and Water Assessment Tool (SWAT) and MIKE HYDRO Basin. The SWAT model is a physical model that requires the introduction of a series of spatiotemporal data to predict the impact of human interventions (e.g., changes in agriculture practices and the addition of chemical fertilizers) on hydrological and chemical processes in river basins [8]. MIKE HYDRO Basin was developed by the Danish Hydraulic Institute [9] and is used as a decision support tool for integrating water resources management and planning in a river basin. This software provides comprehensive hydrological modeling for basin-scale solutions and accommodates a basin-wide representation of water availability and water demand [10].

In this context, the objective of this study is to evaluate the present and future agricultural sustainability of the Sabor River basin through the development of a hydrological and water allocation model (an integration of SWAT and MIKE HYDRO Basin). The specific aims of this study are: (i) to determine whether the quantity of water produced in the basin between 1960 and 2008 was sufficient to satisfy the requirements of the crops and the domestic consumption; and (ii) to determine the quantity of water that will be necessary: (a) to irrigate the total irrigable area that exists in the basin (the first scenario); and (b) with a 24% decrease in the resident population and a 29% increase in the olive grove area for 2060 (the second scenario).

## 2. Materials and Methods

### 2.1. Study Area

The present work was carried out in an agro-forested basin in the northeast of Portugal. The Sabor River basin is a tributary of the Douro River (Figure 1). The main watercourse, with its headwaters located in Spain, is 212.6 km long, and the catchment has an area of approximately 3834.5 km^2^ (3170.7 km^2^ for the Portuguese part). The average annual precipitation and temperature in the region for the period 1957–2008 were 730 mm∙year^−1^ and 12.5 °C, respectively [11]. The weather is typical Mediterranean, characterized by high temperatures and low air humidity in summer [12] and high annual and interannual differences in rainfall quantity and intensity in the autumn and winter months [13]. In the Sabor River basin, the elevation ranges between 88 and 1464 m (Figure 1a). Slopes of less than 15% occupy 58% of the basin according to the Directorate-General of Territory [14]. The soils are dominated by Lithosols (87% of the catchment area), but Cambisols (7.4%), Alisols (3%), Anthrosols (1.4%), and Fluvisols (0.7%) are also present [15]. According to the Corine Land Cover 1990 data published by the European Environmental Agency [16], agricultural area occupies 59%, semi-natural area occupies 31%, and forested area occupies 9% of the total basin area. The remaining 1% is covered by artificial areas and water bodies. According to Statistics Portugal, the irrigation of crop land consumes, on average, 6733 m^3^ of water per hectare per year [17], and, in the Trás-os-Montes region, 70% of this water is groundwater [18].

According to the 2011 census, the population density in the municipalities of the Sabor River basin was on average 20 inhabitants·km^−2^ (Table 1) [19]. The highest population density (greater than 21 inhabitants·km^−2^) was located in West of the basin, in the municipalities Bragança, Macedo de Cavaleiros, Vila Flor, and Carrazeda de Ansiães (Figure 1b). However, from 1960 to 2011, the population density decreased to 14 inhabitants·km^−2^. In this period, the resident population was 176,947 and 103,336 inhabitants, respectively (Table 1). In this period, the average of the decrease was 42% with values that ranged between 63% in Vimioso and 6% in Bragança. In terms of domestic water consumption, the average of all municipalities was 50 L·day^−1^ with values that ranged between 39 L·day^−1^ in Miranda do Douro and 74 L·day^−1^ in Mogadouro (Table 1).

### 2.2. Identification of Irrigable Crops

The Land Use and Land Cover (LULC) of Continental Portugal for 2007 (COS2007) was used to calculate the irrigable area in the Sabor River Basin. These land use data were selected because they have a scale of 1:25,000 and a minimum cartographic unit of 1 hectare [21]. The COS2007 is a European initiative, included in the Coordination of Information on the Environment (CORINE) program, in which one of the main components is cartographic information on land use and cover [22]. The nomenclature of COS2007 is hierarchical, with five levels and 193 LULC classes at the most detailed level. We identified seven irrigable crops: maize, potato, olive groves, fruit trees, horticulture, vineyards, and forages (Table 2). The crops attributed to each class (level five) of COS2007 and the respective areas can be found in Appendix A. The class Temporary irrigated land associated with olive groves, fruit trees, and agroforestry systems was classified as 50% maize and 50% potato. The class Agriculture with natural vegetation and semi-natural areas was also classified as 50% maize and 50% potato, but only 25% of the area of this class is considered to be irrigable area according to the description of [21]. The class Complex cultivation patterns was classified as 100% horticulture, but only 75% of the area of this class is considered to be irrigable area according to the description of [21]. The classes Olives groves and Olives groves with orchards or vineyards were classified as 100% olives, and the Orchard classes, such as almond, chestnut, citrus, and fresh fruit, were classified as 100% orchards. The classes Vineyards and Vineyards with olive groves or orchards were classified as 100% vineyards and the class Permanent pastures was classified as 100% forages. This classification shows that, in the study area, the dominant form of agriculture is olive groves, with 322 km^2^ distributed over the centre and southeast of the basin (Table 2). Fruit trees are the second most dominant form of agriculture, with 130 km^2^ of area located essentially in the north of the basin.

However, according to the agricultural census of 1999 [3], only a small percentage of the irrigable area is, in fact, irrigated. The irrigable area is defined as the maximum crop area that is available for irrigation, and the irrigated area is defined as the area that is irrigated at least once a year. On this basis, the crops with a major percentage of the irrigated area were maize and potato with 90% and 80%, respectively, compared to forages, which had only 5% of the irrigated area (Table 2).

### 2.3. Conceptual Framework

A conceptual framework was developed to model the hydrology and water allocation in the Sabor River basin (Figure 2). For that, we used a modelling method that integrates the software packages SWAT and MIKE HYDRO Basin. In SWAT, a detailed database of the catchment was developed for hydrological analysis and modelling. The database was composed of the following data: digital elevation model, land use and land cover, soil characteristics, and hydro-meteorological data. A hydrological model was constructed using all these data. Then, this model was calibrated using data from the period 1960–1999 in the SWAT-CUP (Calibration and Uncertainty Procedures) software and validated using data from the period 2000–2008. After that, the hydrological model was used to simulate the hydrologic processes in the Sabor River basin during the period 1960–2008 and some output data were input into MIKE HYDRO Basin. The output data were: drainage network, basin and sub-basin delimitation, surface runoff, aquifer recharge, rainfall, and potential evapotranspiration at sub-basin scale data. Then, the data on the water that was used for domestic consumption and irrigation, at the sub-basin scale, were also input into MIKE HYDRO Basin.

In MIKE HYDRO Basin, a first phase simulated the water that was used for domestic consumption and irrigation between 1960 and 2008. This period was called the historical view. In a second phase, two scenarios were created: the first was used to simulate the water that was used for domestic consumption between 1960 and 2008 assuming, with respect to irrigation, that the entire irrigable area identified in COS2007 was irrigated [21]. The second scenario was used to simulate the expansion of the olive grove area and the decrease in the resident population according to the projection for 2060. The simulation for the two scenarios was performed with surface runoff, aquifer recharge, precipitation, and reference evapotranspiration data for the period 1960–2008. All simulations were performed on a daily time step between 1960 and 2008 in both SWAT and MIKE HYDRO Basin.

The historical view and the two scenarios were analyzed in order to evaluate the impacts of water management, agricultural practices, and the demographic trend on water allocation in the Sabor River basin.

### 2.4. SWAT Hydrological Model and Calibration

The SWAT model has been applied all over the world, to watersheds of different sizes, and to assess the effects of climate and land use changes on water balance, water quantity, nutrient exportation, and soil erosion [23]. This software is based on a mathematical model developed by the United States Department of Agriculture (USDA) [23,24]. SWAT runs in the Geographic Information System (GIS) software ArcMap (www.esri.com) developed by the Environmental Systems Research Institute (ESRI; Redlands, NY, United States) and widely applied to environmental studies [25,26,27,28,29,30,31,32,33,34]. In this study, the 2012 version of SWAT was used to model the hydrologic processes in the Sabor River basin. Into this software, we inserted digital elevation model, land-cover map, soil-type map, and meteorological data. The sources and scale of the data are summarized in Table 3.

The Trás-os-Montes and Alto Douro digital elevation models, which are available from the Directorate-General of Territory (http://www.dgterritorio.pt/), were used to delimit 37 sub-basins and define the drainage network. The Corine Land Cover (CLC) of 1990, 2000, and 2006, which are available from the European Environment Agency [16], were used as a land cover map. The CLC 1990, 2000, and 2006 data were used to model the land cover between 1960 and 1999, between 2000 and 2005, and between 2006 and 2009, respectively. The Trás-os-Montes and Alto Douro soil type data, which are available from the Directorate-General of the Territory [14], were used as a soil type map. The precipitation data from 11 meteorological stations and the meteorological data from the Folgares station (06N/01C) (Figure 1c), which are available from the National System of Water Resources Information [35], were inserted into the SWAT model. The Thiessen polygon method was selected to interpolate the precipitation data from the basin. The meteorological data from Folgares station was used in the Penman–Monteith equation. More details on the input data can be found in Santos et al. [36]. With all these data, the SWAT hydrological model of the Sabor River basin was executed for the period 1960–1999. Then, the hydrological model was calibrated and validated with streamflow data that were collected at the Quinta das Laranjeiras hydrometric stations between 1960 and 1999 and between 2000 and 2008, respectively. The calibrated model’s parameters are presented in Santos et al. [36]. A complete description of the SWAT model and its underlying theory can be found in [37].

The model’s calibration was performed using the SWAT-CUP 2012 software on a daily time step [24,37]. The model’s performance (defined as the goodness-of-fit between the observed and the calibrated streamflow) was evaluated using the coefficient of determination (R^2^), the Nash–Sutcliffe coefficient (NSE), the ratio of the root-mean-square error to the standard deviation of measured data (RSR), and the percent of bias (PBIAS) [38]. These performance indicators are widely used in hydrologic and statistical modeling of environmental data [39,40,41,42,43]. A model’s performance for streamflow is considered to be satisfactory when R^2^ and NSE are greater than 0.5, RSR is less than 0.7, and PBIAS ranges between less than ±25%. The calibrated and the validated SWAT hydrological model was then run for the period 1960–2008.

### 2.5. MIKE HYDRO Basin Model

MIKE HYDRO Basin has been applied to a large number of watersheds around the world with different climates and hydrological regimes [44,45,46,47]. MIKE HYDRO Basin has been used to simulate the components of the rainfall–runoff process (overland flow, interflow, and baseflow) [45,48], predict the impacts of climatic changes on habitat conservation of the endangered pearl mussel [49], estimate the nutrient loads from diffuse and point sources in river water [50,51], estimate the concentration of phosphorus in a river due to recurrent wildfires [52], and estimate the water availability for a population in climate change scenarios [47,53].

The MIKE HYDRO Basin model is composed of catchment, river network, water user, and reservoir operation, including hydropower, elements. The river network includes branches, river links, river nodes, priority nodes, and routing. The water users include a regular water user and an irrigation water user. Into the regular water user module was input the amount of water used for domestic consumption, which was calculated using the resident population and the water consumption per municipality. Data on the resident population per year between 1960 and 2008 are available from the National Statistics Institute (INE) [19]. Data on water consumption between 1995 and 2009 per municipality (Table 1) are available from PORDATA [20]. These data are expressed in L·day^−1^·inhabitant^−1^, and we used the average daily consumption per municipality to calculate the water consumption (Table 1).

The irrigation water user represents an irrigation area that comprises one or more irrigated crops and the total water requirements for the fields. The water requirements can be extracted from one or more sources, e.g., river nodes and/or reservoirs, according to specific allocation rules. The crop water requirement is defined as the amount of water that is required to compensate for the evapotranspiration loss (ET) from the cropped field and depends on the local climate and the crops growing in the fields. It is computed, according to the dual crop coefficient model approach, as follows [54]:(1)$$ET_{c}=\left(K_{cb}+K_{e}\right)ET_{0}$$where $ET_{c}$ is the crop or maximum evapotranspiration (mm·day^−1^) under the potential growing conditions with no soil water shortage stress, $K_{cb}$ is the basal crop coefficient that describes crop transpiration, $K_{e}$ is the soil water evaporation coefficient that describes soil evaporation, and $ET_{0}$ is the reference evapotranspiration value, which is calculated using the Penman–Monteith equation.

The irrigation module follows the methodology proposed by the Food and Agriculture Organization of the United Nations FAO-56 Dual Crop Coefficient, which concerns the parameters to be assigned to the crop model [54]. The irrigation module includes the crop data (crop-specific details), the irrigation method (how and when irrigation occurs), and the soil type. The crop data comprise a set of parameters, namely: crop stages, sowing date, basal crop coefficient (*K_cb_*), root depth, maximum height, and depletion fraction. Crop stages are divided into an initial stage, a development stage, a mid-season stage, and a late-season stage, and the periods are given in days. The basal crop coefficient (*K_cb_*) is defined as the crop evapotranspiration over the reference evapotranspiration (*ET_c_/ET*_0_) when the soil surface is dry but transpiration is occurring at the potential rate, i.e., water is not limiting transpiration [54]. The root depth is specific to the initial and middle stages, and the depletion fraction is an average fraction of the total available soil water that can be depleted from the root zone before moisture stress. The parameters’ values are provided by the FAO tables [54] and are shown for the crops identified in the Sabor River basin in Table 4.

The irrigation methods considered were the sprinkler method and the gravity irrigation and drip irrigation method. The sprinkler method was used for the maize, potato, forages, and horticulture crops, and the drip irrigation method was used for the vineyard, olive grove, and fruit tree crops. For each method, the spray loss and wetting fraction were defined. The spray loss corresponds to the fraction of the irrigation water that is evaporated before the water reaches the soil surface. The wetting fraction determines the fraction of the field surface that is being wetted during irrigation. The wetting fraction is an important factor for determining how much irrigation is required before the surface soil’s storage is filled and, hence, when the root zone starts to fill. For the sprinkler method crops, we adopted a spray loss of 0.25 and a wetting fraction of 1. For the drip irrigation method crops, we adopted a value of 0.1 for both the spray loss and the wetting fraction, except for the fruit tree crops, for which we adopted a spray loss of 0.15 and a wetting fraction of 0.25.

According to the 2009–2011 National Agricultural Census [18], in the Trás-os-Montes region, 70% of the water that is used for irrigation comes from groundwater sources, and the remainder comes from surface flow. Since the Sabor River basin is located in the Trás-os-Montes region, we applied these percentages to the water sources.

The dominant soil type in the basin is sandy loam (63.4% sand, 22.8% silt, and 14.8% clay). The sandy loam parameters and their respective values were: field capacity (0.2), wilting point (0.1), initial soil moisture (0.1), depth of evaporable layer (0.1), and porosity (0.45). The parameter values were obtained from the FAO tables [54].

### 2.6. The MIKE HYDRO Basin Model Input and Output Data

The MIKE HYDRO Basin model is composed of 37 sub-basins and rivers. For each sub-basin, we added a regular water user and an irrigation user (Figure 3a). SWAT was used to delimit the 37 sub-basins and rivers and obtain a time series of surface runoff and aquifer recharge. Into each regular water user was input the water consumption, and into each irrigation user was input a time series of precipitation and the reference evapotranspiration. All time series were input into each of the sub-basins, on a daily basis, between 1960 and 2008. Figure 3a shows each sub-basin’s number, and Figure 3b shows the name of each river.

The output of the model is presented for each catchment, node, and river of the basin. The values extracted from the nodes were the quantities of the streamflow and water used for domestic consumption and irrigation. For irrigation, the values of irrigation demand, applied irrigation, and deficit irrigation per sub-basin are available. The irrigation demand (expressed in m^3^·km^−2^ per crop) was compared with a reference irrigation allocation. The irrigation demand, applied irrigation, and deficit irrigation (expressed in m^3^ for all crops for each sub-basin) were used to analyze the amount of water consumed, the amount of water required to irrigate the crops, and the water deficit of the crops, respectively. The streamflow, the water used for domestic consumption, the applied irrigation, and the deficit irrigation were expressed in m^3^·s^−1^ in order to construct distribution maps.

The applied irrigation in the basin was compared with a reference irrigation allocation from the Directorate-General for Agriculture and Rural Development (DGARD) [55] and Statistics Portugal (National Statistics Institute, INE) [17] for each crop. The reference irrigation allocation from DGARD is for the Northern and Central Interior Region of Portugal, and the reference irrigation allocation from the INE is for Portugal. The DGARD provides a reference irrigation allocation for groundwater and for different surface irrigation methods (sprinkler, micro-sprinkler, cannon, and drip irrigation). In this study, the average of the sprinkler, micro-sprinkler, and cannon irrigation methods was used for the maize, potato, forages, and horticulture crops, and drip irrigation was used for the vineyard, olive grove, and fruit tree crops. The reference irrigation allocation from DGARD and the INE for the olive grove, fruit tree, forages, and horticulture crops was the average of the cultures that occurred in the study area. For example, the reference irrigation allocation of horticulture is the average of the tomato, onion, carrot, melon, and pea cultures. The calculations for the reference irrigation allocation are presented in the Appendix A.

### 2.7. Historical View and Evolutionary Scenarios

The historical view modelled the domestic water consumption and irrigation in the Sabor River basin between 1960 and 2008. The simulation used the percentage of irrigated area per crop shown in Table 2. The values of irrigated area per sub-basin can be found in Appendix A. The areas were input into each sub-basin in the irrigation module of MIKE HYDRO Basin.

In the first scenario, we modelled the domestic water consumption with a time series of surface runoff, aquifer recharge, precipitation, and reference evapotranspiration data for the period 1960–2008. In the irrigation module, we assumed that the entire irrigable area identified in COS2007 was irrigated [21] (Table 2). The area of each irrigable crop per sub-basin can be found in the Appendix A.

In the second scenario, we modelled both a demographic projection and an irrigation projection for 2060 with a time series of surface runoff, aquifer recharge, precipitation, and reference evapotranspiration data for the period 1960–2008. The demographic projection was based on a projection scenario in a technical report on the resident population between 2012 and 2060 in the north zone and central region [56]. The projection estimates a 24% decrease in the resident population by 2060. Figure 4a shows the gradual reduction of the resident population from 2012 to 2060 for all municipalities in the Sabor River basin. The projection estimates a reduction of 25,056 inhabitants in municipalities of the basin by 2060, which means five less inhabitant·km^2^. The municipalities with the largest projected reduction in population are Bragança and Macedo de Cavaleiros with 8495 and 3793 inhabitants, respectively. In the other municipalities, the reduction in population is projected to range between 910 and 2296 inhabitants.

The irrigation module was used to model the expansion of the olive grove area. According to the agricultural census [4,5,6] and the land use and land cover data for 2007 and 2015 [57], from 1989 to 2015, the olive grove area increased by 86 km^2^ in the Sabor River basin (Figure 4b). The olive grove area’s expansion was expressed using a robust linear regression (R^2^ = 0.97) and used to provide a projection on the olive grove area in 2060 (Figure 4b). The area is projected to expand by 172 km^2^, which represents a 29% increase.

A projection on the olive grove area’s spatial distribution in 2060 was made according to land uses that had been converted into olive groves between COS2007 and COS2015. The main land uses in COS2007 that had been converted into olive groves in COS2015 were non-irrigated arable land, moors and heathland, transitional woodland–shrub, and eucalyptus forest. From these land uses, we randomly selected 29% of the area and assumed that, in 2060, it will be olive groves. The olive grove area’s expansion per sub-basin can be found in the Appendix A.

## 3. Results

### 3.1. Calibration and Validation of the Streamflow

SWAT was used to simulate the daily streamflow in the Sabor River basin over 49 years (1960–2008). The streamflow was first calibrated to a 39-year period (1960–1999) and then validated to a 9-year period (2000–2008). The goodness-of-fit indicators from Table 5 indicate that the model’s performance is satisfactory according to the model evaluation guidelines by Moriasi et al. [38].

The streamflow’s calibration shows a satisfactory performance with R^2^, RSR, and NS values of 0.63, 0.62, and 0.62, respectively, and a very good PBIAS (2.7%) [38]. The same goodness-of-fit indicators were obtained for the validation, with a very good R^2^ (0.8) and satisfactory RSR, NS, and PBIAS values (0.63, 0.61, and −24%, respectively) (Table 5). The negative PBIAS value indicates a model overestimation bias [38]. Figure 5a,b illustrate the calibrated and validated streamflow, respectively. As shown in the figures, there is good agreement between the observed and the simulated streamflow. The parameters and their respective values that resulted from the calibration in SWAT-CUP can be found in Santos et al. [36].

Figure 6a,b shows the average values of streamflow and precipitation, respectively. The streamflow is represented at the sub-basin scale and for the period 1960–2008. The streamflow over most of the basin’s surface is less than 40 × 10^6^ m^3^ of water. The greatest values were recorded in the Azibo river (94 × 10^6^ m^3^), followed by the Vale of Moinhos, upstream of the Baixo Sabor and Zacarias rivers, with values that ranged between 41 and 54 × 10^6^ m^3^ (Figure 3b and Figure 6a). The obtained precipitation values are based on the spatial interpolation of data from the meteorological stations used in SWAT to construct the hydrological model. The interpolation shows that the highest values were recorded in the sub-basins located in the west of the basin, with values that ranged between 800 and 1137 mm (Figure 6b).

### 3.2. Irrigation Demand

The simulation of irrigation demand was performed on a daily basis, at the sub-catchment scale, between 1960 and 2008 in the Sabor River basin. Table 6 presents the results of the irrigation demand simulation by the MIKE HYDRO Basin irrigation scheme and the reference irrigation allocation for each crop. The irrigation demand represents the average annual water consumption of each crop in the basin. The maize, potato, and fruit tree crops had the highest water resource requirements (an average allocation higher than 5300 m^3^·ha^−1^). The horticulture crop also had a high water allocation requirement (approximately 4500 m^3^·ha^−1^), followed by forages (3475 m^3^·ha^−1^). Vineyards and olive groves had the lowest water resource requirements of the analyzed crops (2436 and 2783 m^3^·ha^−1^, respectively).

These results fall within the range of irrigation allocation reference values given by the Directorate-General for Agriculture and Rural Development for the Northern and Central Interior Region of Portugal [55] and Statistics Portugal for Portugal [17] (Table 6). The irrigation allocation reference values, per crop and irrigation method, can be found in the Appendix A. The irrigation demand for potato and olive groves falls into the range of reference values. The irrigation demand for fruit trees and horticulture is close to the minimum reference value (5817 and 4514 m^3^·ha^−1^, respectively). The irrigation demand for forages (3475 m^3^∙ha^−1^) was the farthest from the minimum reference value (4213 m^3^·ha^−1^). The irrigation demand for vineyards is close to the maximum reference value (2302 m^3^·ha^−1^).

### 3.3. Historical View and Evolutionary Scenarios of Domestic Consumption and Irrigation

Table 7 shows the total water demand, the water used, and the water demand deficit for domestic consumption in the Sabor River basin. The values are expressed as the annual average in the period 1960–2008 for the historical view and for the two scenarios (the increase in irrigated area and the 29% increase in Olive grove area in 2060). The total water demand in the historical view, and in the increase in irrigated area scenario, was 1372 × 10^3^ m^3^. The total water demand in the scenario of a 29% increase in the Olive grove area in 2060 was 1045 × 10^3^ m^3^. The water used for domestic consumption in the historical view was 950 × 10^3^ m^3^ and was decreased in both the scenario of an increase in the irrigated area and the scenario of a 29% increase in the Olive grove area in 2060 (866 and 659 × 10^3^ m^3^, respectively). The water demand deficit in the historical view was 422 × 10^3^ m^3^ (31%). It increased in the scenario of an increase in the irrigated area (506 × 10^3^ m^3^), and decreased in the 2060 projection scenario (385 × 10^3^ m^3^). In both scenarios, the water deficit was 37%.

In the historical view, the annual irrigation demand for all crops was calculated to be 94 × 10^6^ m^3^ for 181.6 km^2^ of the irrigated area (Table 8). Fruit trees were found to have the highest irrigation demand (53.7 × 10^6^ m^3^), followed by maize and potato (14.1 × 10^6^ m^3^ and 12.8 × 10^6^ m^3^, respectively). Forages (0.03 × 10^6^ m^3^) and vineyards (0.75 × 10^6^ m^3^) were found to have the lowest irrigation demand. The irrigation demand for olive groves and horticulture was 6.2 × 10^6^ m^3^ and 6.6 × 10^6^ m^3^, respectively. However, only 28.8 × 10^6^ m^3^ of water was found to have been applied, which represents an irrigation deficit of 65.4 × 10^6^ m^3^ for all crops. The irrigation deficit ranges between 50% and 68% of the plants’ water requirements. As expected, the highest irrigation deficit was calculated for the crops with the highest irrigation demand, such as fruit trees, maize, and potato (37.6, 10, and 9 × 10^6^ m^3^, respectively).

In the first scenario, we assumed that the entire irrigable area was irrigated (586.8 km^2^). The aim is to determine the quantity of water that is necessary to satisfy each crop’s water demand. The results show that 237 × 10^6^ m^3^ of annual irrigation is needed to satisfy the water requirements of all of the crops (Table 8). The highest irrigation demand was calculated for olive groves and fruit trees (91.22 × 10^6^ m^3^ and 81.33 × 10^6^ m^3^, respectively). Forages were found to have the lowest irrigation demand (0.64 × 10^6^ m^3^). With respect to the other crops, the irrigation demand ranged between 11 and 18 × 10^6^ m^3^. However, only 54.7 × 10^6^ m^3^ of irrigation was found to have been applied, which represents an irrigation deficit of 182.5 × 10^6^ m^3^ for all crops. The irrigation deficit ranges between 55.4% and 78% of the plants’ water requirements. As expected, the highest irrigation deficit was calculated for the crops with the highest irrigation demand, such as olive groves and fruit trees (69.3 and 63.6 × 10^6^ m^3^, respectively).

In the second scenario, we projected a 140.1 km^2^ increase in the olive grove area for 2060. In this scenario, the irrigated area of all other crops is the same as that in the historical view period (1960–2008). The results show that 235.3 × 10^6^ m^3^ of annual irrigation is needed to satisfy the water requirements of all of the crops (Table 8). For olive groves, only 33 × 10^6^ m^3^ of irrigation was found to have been applied, which represents an irrigation deficit of 107 × 10^6^ m^3^. The irrigation deficit was 70.6% of the crops’ total water requirement. For all other crops, the irrigation demand, applied irrigation, and deficit irrigation were close to the values obtained for the historical view.

### 3.4. Spatial Distribution of Water Allocation

Figure 7 shows the spatial distribution of the water used for domestic consumption and the water demand deficit per sub-basin of the Sabor River basin. The values are the annual average between 1960 and 2008 for the historical view and for the two scenarios (an increase in the irrigated area and a projection to 2060). In the historical view (between 1960 and 2008) (Figure 7a,b), most of the basin’s surface consumed between 22 and 58 × 10^3^ m^3^ of water. The greatest water consumption was recorded in the Azibo river, the Vilariça river, and downstream of the Baixo Sabor river, with values that ranged between 62 and 95 × 10^3^ m^3^ (Figure 7a). In terms of the water demand deficit, the highest values were recorded in the Azibo and Vilariça rivers (44 × 10^3^ m^3^ and 46 × 10^3^ m^3^, respectively) (Figure 7b).

In the scenario of an increase in the irrigated area (Figure 7c,d), most of the basin’s surface is able to consume between 20 and 55 × 10^3^ m^3^ of water, and the highest consumption of water occurs in the Azibo river (87 × 10^3^ m^3^) (Figure 7c). The highest water demand deficits occur in the Azibo, Vale de Moinhos, and Vilariça rivers with values that range between 41 and 54 × 10^3^ m^3^ (Figure 7d).

In the projection to 2060 scenario (Figure 7e,f), most of the basin’s surface is able to consume between 21 and 40 × 10^3^ m^3^ of water, and the highest water consumption occurs in the Azibo river and upstream of the Douro Superior river (66 × 10^3^ m^3^ and 46 × 10^3^ m^3^, respectively) (Figure 7e). The water demand deficit is 20 × 10^3^ m^3^ of water for most of the basin’s surface. The highest deficits occur in the Azibo, Vale de Moinhos, Maças, and Vilariça rivers with values that range between 20 and 41 (Figure 7f).

Figure 8 shows the spatial distribution of the applied irrigation and the irrigation deficit per sub-basin of the Sabor River basin. The values are expressed as the annual average between 1960 and 2008 for the historical view and for the two scenarios (an increase in the irrigated area and a projection to 2060). In the historical view, the highest amounts of applied irrigation (Figure 8a) and irrigation deficits (Figure 8b) were found in rivers situated in the west of the basin, more precisely in the Sabor Superior river, the Vale of Moinhos, the Azibo river, the Zacarias river, and upstream of the Angueira river (Figure 3b). In these rivers, the quantity of applied irrigation ranged between 1.6 and 3.4 × 10^6^ m^3^ of water and the irrigation deficit between 4.8 and 7.8 × 10^6^ m^3^ of water.

In the scenario of an increase in the irrigated area (Figure 8c,d), the quantities of applied irrigation ranged between 0.01 and 3.9 × 10^6^ m^3^ of water, except for in the Azibo river (6.6 × 10^6^ m^3^ of water) (Figure 8c). The irrigation deficit was highest in the Vale of Moinhos, the Azibo river, the Zacarias river, the Vilariça river, and downstream of the Baixo Sabor river, with values that ranged between 10.3 and 22 × 10^6^ m^3^ of water (Figure 8d).

In the projection to 2060 scenario (Figure 8e,f), the quantities of applied irrigation ranged between 0.01 and 4.1 × 10^6^ m^3^ of water, except for in the Azibo river (6.5 × 10^6^ m^3^ of water) (Figure 8e). The irrigation deficit is highest in the Vale of Moinhos, the Azibo river, the Zacarias river, the Vilariça river, and downstream of the Baixo Sabor river, with values that range between 12.7 and 24 × 10^6^ m^3^ of water (Figure 8f).

## 4. Discussion

### 4.1. Historical View of Water Allocation

The annual average quantity of water that left the Sabor River basin during the period 1960–2008 was 629 × 10^6^ m^3^ (or 20 m^3^·s^−1^). The domestic consumption of water is, on average, 0.15% of the total amount of water in the basin, and 0.22% of the total amount of water in the basin is necessary to meet all domestic water consumption needs (Table 7 and Figure 6a). However, irrigation consumes an annual average of 5% of the total amount of water in the basin, and 15% of the total amount of water in the basin is necessary to satisfy all irrigation demand. (Table 8 and Figure 6a). The water used for domestic consumption was 950 × 10^3^ m^3^, and the water deficit was 422 × 10^3^ m^3^ (Table 7). The annual irrigation demand was 94 × 10^6^ m^3^, and the water demand deficit was 65.4 × 10^6^ m^3^ (Table 8). The water demand deficit for domestic consumption was 31%, and the water demand deficit for irrigation was 69.5%. This high water demand deficit is due to the low flow period in spring and summer and the high variation in river flow during the year. Figure 9 shows the annual variation in total irrigation demand, applied irrigation, streamflow, precipitation, and evapotranspiration for the period 1960–2008. Critical periods of irrigation deficit occur in July and August because the total irrigation demand is higher than the streamflow. Between March and October, except for the critical periods, the streamflow is higher than the total irrigation demand. However, the applied irrigation is less than the total irrigation demand. This happened because, in most years, the river flow was low, and could not supply the irrigation demand. The low streamflow is a consequence of the low precipitation and high evaporation that Mediterranean countries experience during the spring and summer [10,45,58].

### 4.2. Evolutionary Scenarios

In the scenario of an increase in the irrigated area, we assumed that the total area of each crop was irrigated. In this scenario, we found that 237.2 × 10^6^ m^3^ of water will be necessary to irrigate 586.8 km^2^ of crop area. If we consider the same streamflow between 1960 and 2008, only 54.7 × 10^6^ m^3^ of water is necessary for irrigation. The deficit will be 182.5 × 10^6^ m^3^ of water, which represents 76.9% of the total irrigation demand (Table 8). The highest irrigation demand was calculated for olive groves (91.22 × 10^6^ m^3^ of water), and fruit trees (81.33 × 10^6^ m^3^ of water). The water requirement for these crops is high because they occupy a large irrigated area (322 and 131 km^2^, respectively). These crops represent 77% of the irrigable area of the basin (Table 8). Moreover, the water deficit is higher: approximately 69.3 × 10^6^ m^3^ and 64 × 10^6^ m^3^ of water for olive groves and fruit trees, respectively. The quantity of water used for domestic consumption is 866 × 10^3^ m^3^, and the water deficit is 506 × 10^3^ m^3^, which represents 37% of the total water demand. The results show that, in this scenario, the increase in the water demand deficit of 84 × 10^3^ m^3^ for domestic consumption, and of 117.1 × 10^6^ m^3^ for irrigation, when compared to the historical view, is due to the increase in the irrigation area and the lower availability of water, especially in spring and summer.

In the scenario of a projection to 2060, the expansion of the olive grove area represents a 172 km^2^ increase in this crop’s irrigable area. The scenario was projected with the streamflow for the period 1960–2008 and the irrigated area for the other crops that was reported in the historical view. In this scenario, 140.1 × 10^6^ m^3^ of water will be necessary to irrigate 494 km^2^ of olive grove area, but only 33.1 × 10^6^ m^3^ of water will be applied, which represents a deficit of 107 × 10^6^ m^3^ of water. A total of 235.3 × 10^6^ m^3^ of water will be necessary to irrigate the entire area (approximately 653 km^2^), but only 55.2 × 10^6^ m^3^ of water will be applied to irrigation. The deficit will be 180 × 10^6^ m^3^ of water, which represents 76.6% of the total irrigation demand (Table 8). The 24% decrease in the resident population by 2060 represents a reduction of 25,056 inhabitants and five less inhabitants·km^2^ in the municipalities in the basin. This population reduction will produce a 327 × 10^3^ m^3^ decrease in total water demand. However, a water demand deficit of 385 × 10^3^ m^3^ will remain because of the increase in the olive grove irrigation area.

### 4.3. Spatial Distribution

The spatial distribution of the water used for domestic consumption and irrigation in the historical view and in both scenarios shows that the highest consumption of water occurred in the west of the basin (Figure 7). The reason for this is that the basin’s western region has the highest population density (>20 inhabitants·km^−2^), which is responsible for the domestic water consumption (Figure 1b), and is a large irrigable area where olive groves and fruit trees are predominant (Figure 1c). Moreover, a large streamflow is located in the west of the basin with values that range between 41 and 94 × 10^6^ m^3^. This streamflow is due to the large amount of precipitation (between 800 and 1137 mm) that falls in this part of the basin.

In the historical view, most of the basin’s surface consumed between 22 and 58 × 10^3^ m^3^ of water. In the scenario of an increase in the irrigated area, the highest consumption ranged between 20 and 55 × 10^3^ m^3^ of water. In the scenario of a projection to 2060, the highest consumption ranged between 21 and 40 × 10^3^ m^3^ of water (Figure 7). The highest consumption of water was registered in the Azibo river (95 × 10^3^ m^3^ of water in the historical view, 87 × 10^3^ m^3^ of water in the scenario of an increase in the irrigated area, and 66 × 10^3^ m^3^ of water in the scenario of a projection to 2060). In the west of the basin, the highest irrigation deficits ranged between 4.8 and 7.8 × 10^6^ m^3^ of water in the historical view, between 10.3 and 22 × 10^6^ m^3^ of water in the scenario of an increase in the irrigated area, and between 12.7 and 24 × 10^6^ m^3^ of water in the scenario of a projection to 2060. In other words, the irrigation deficit increased in both scenarios when compared with the historical view.

Transverse barriers play an essential role in the storage of water for irrigation. The Sabor River basin has at least 129 transverse barriers, either dams or weirs [59] (Figure 10). Figure 10 shows that the transversal barriers are mainly located in the Sabor Superior river, the Angueira river, the Maças river, the Vilariça river, and downstream of the Baixo Sabor river (Figure 3). In these rivers, the streamflow ranges between 20 and 40 × 10^6^ m^3^ and the main crops are olive groves, fruit trees, and horticulture (Figure 1c). These transversal barriers contribute to reducing the water deficit in these rivers. However, in rivers where the water deficit is higher (e.g., Vale of Moinhos, the Azibo river, and the Zacarias river), there is a lack of transversal barriers.

## 5. Conclusions

This study evaluated the present and future agricultural sustainability of the Sabor River basin through the development of a hydrological and water allocation model (an integration of SWAT and MIKE HYDRO Basin). SWAT was used to model the components of the water balance, and MIKE HYDRO Basin was used to simulate the water allocation, irrigation, and domestic water consumption. The results show that the Sabor River basin does not have enough water to fulfil the requirements of the crops. In the historical view, between 1960 and 2008, 1372 × 10^3^ m^3^ of water was found to be necessary for domestic water consumption and 94 × 10^6^ m^3^ of water for irrigation. However, the average annual water demand deficit was 31% for domestic water consumption and 70% for irrigation. Therefore, to fulfil the crop requirements, 422 × 10^3^ m^3^ more water is necessary for domestic consumption and 65.4 × 10^6^ m^3^ more water is necessary for irrigation. The high water demand deficit, especially for irrigation, is due to the variation in river flow that exists during the year and the low flow period in spring and summer, which is exactly when irrigation demand is higher.

In the scenario of an increase in the irrigated area from 181.6 to 586.8 km^2^, the water demand deficit of domestic consumption increased to 37%, and the water demand deficit of irrigation increased to 77%. Therefore, to fulfil the crop requirement, 506 × 10^3^ m^3^ more water is necessary for domestic consumption and 183 × 10^6^ m^3^ more water is necessary for irrigation. The same percentage of water demand deficit was obtained in the scenario of a projection to 2060. In 2060, a 24% decrease in the resident population is expected, which will represent five less inhabitants·km^2^, along with a 29% increase in the irrigation area of olive groves, which will represent an additional 172 km^2^ of area. In this scenario, we found the demand deficit to be 385 × 10^3^ m^3^ of water for domestic consumption, and 106 × 10^6^ m^3^ of water for irrigating the olive grove area. Overall, this work provides a reference to stakeholders and decision-makers on how much water is being consumed in the Sabor River basin and how much water needs to be stored to fulfil the present and future water demand.

## Figures and Tables

**Figure 1 ijerph-16-02419-f001:**
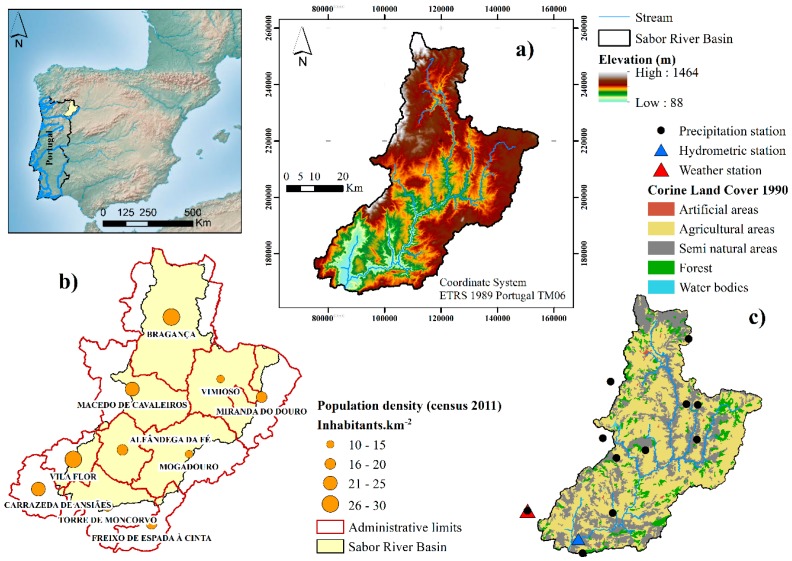
Geographical location of the Sabor River basin. (**a**) Topography and drainage network, (**b**) population density per municipality according to the 2011 census, and (**c**) hydro-meteorological data and the Corine land cover 1990.

**Figure 2 ijerph-16-02419-f002:**
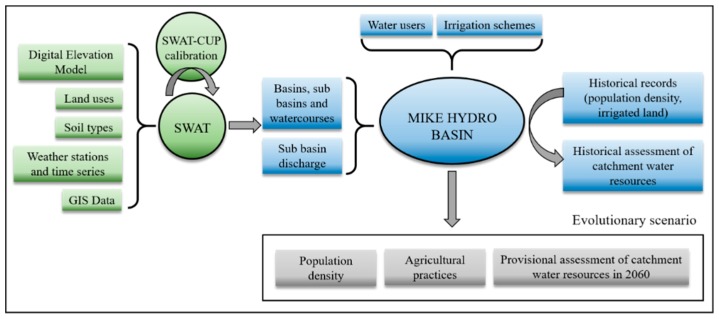
The process and concept for the Soil and Water Assessment Tool (SWAT) and MIKE HYDRO Basin models.

**Figure 3 ijerph-16-02419-f003:**
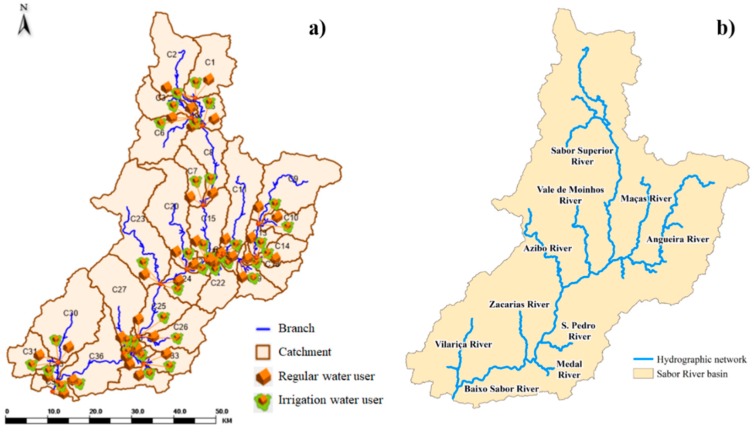
(**a**) The drainage networks, regular water users, and irrigation users at sub-basin scale in MIKE HYDRO Basin and (**b**) all tributaries of the Sabor River basin.

**Figure 4 ijerph-16-02419-f004:**
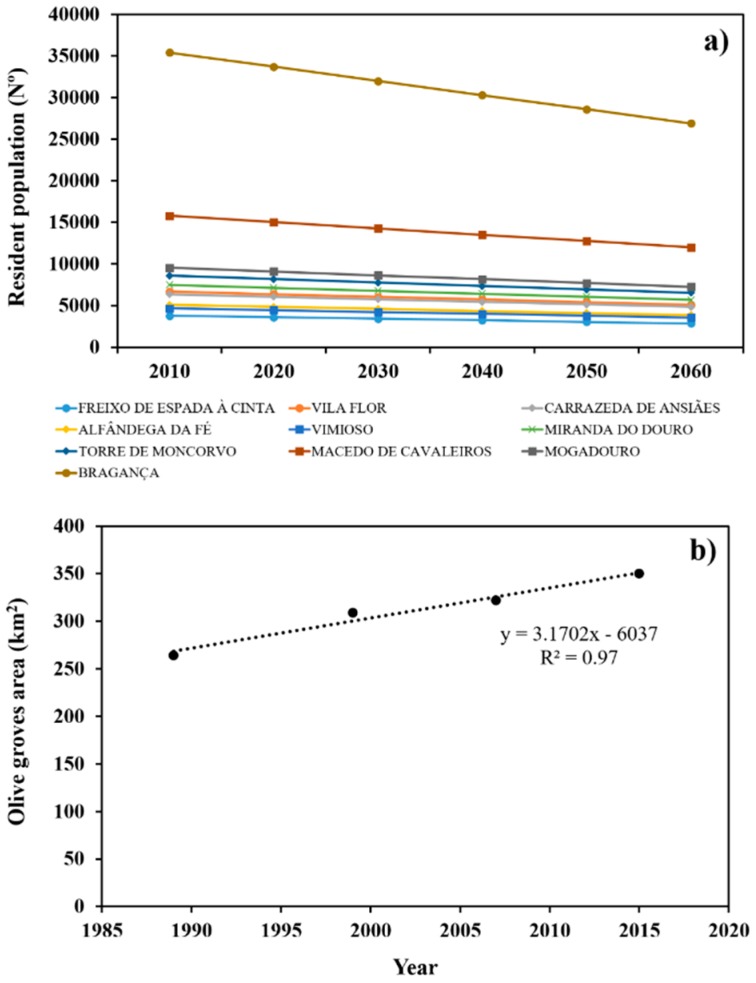
(**a**) Scenario for the resident population’s evolution from 2010 to 2060 for municipalities in the Sabor River basin, and (**b**) Linear regression of the olive grove area between 1989 and 2015. The linear regression was used to make a projection on the olive grove area in 2060. Data on the resident population are from Statistics Portugal [19]. Data on the olive grove area are from the Directorate-General for Agriculture and Rural Development [4,5,6] and the Directorate-General of Territory [14].

**Figure 5 ijerph-16-02419-f005:**
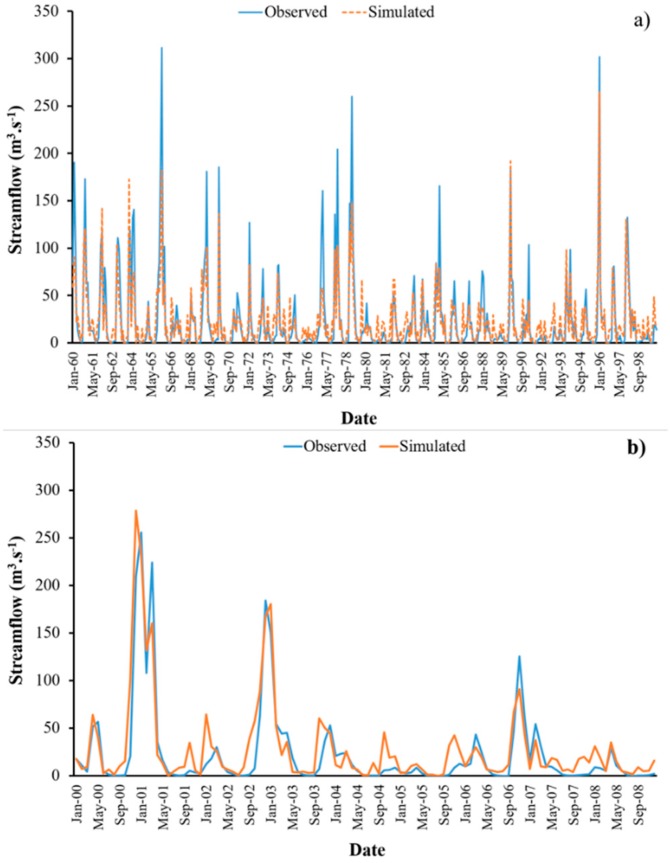
Comparison of observed and simulated streamflow during (**a**) the calibration period (1960–1999); and (**b**) the validation period (2000–2008) in the Sabor River basin. The simulation of the streamflow was executed on a daily basis; however, for the sake of visualization, the results are presented on a monthly basis.

**Figure 6 ijerph-16-02419-f006:**
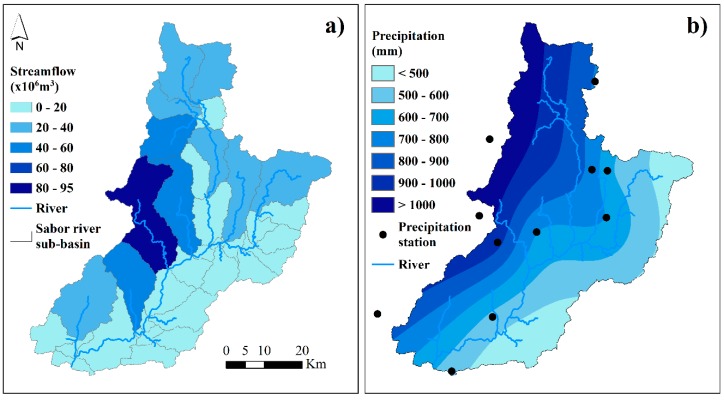
The streamflow and precipitation in the Sabor River basin. (**a**) The streamflow is an annual average value for the period 1960–2008 and is represented at the sub-basin scale. (**b**) The precipitation is based on the spatial interpolation of data from the meteorological stations used in SWAT to construct the hydrological model, with values for the period 1960–2008.

**Figure 7 ijerph-16-02419-f007:**
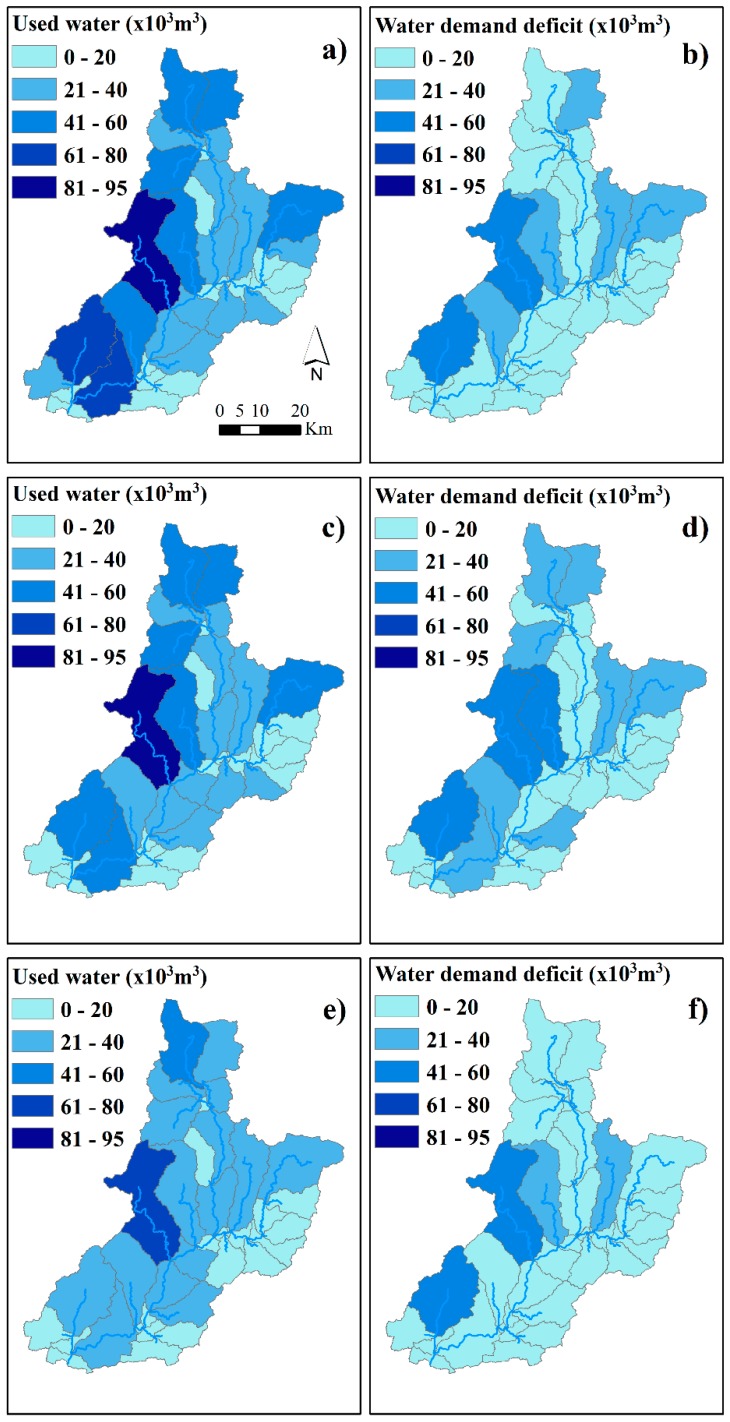
The water used for domestic consumption and the water demand deficit at the sub-basin scale in the Sabor River basin. (**a**) and (**b**) represent the water used for domestic consumption and the water demand deficit in the historical view, respectively. The figures (**c**) and (**d**) represent the data on the increase in the irrigated area, and (**e**) and (**f**) represent the data on the scenario of a projection to 2060.

**Figure 8 ijerph-16-02419-f008:**
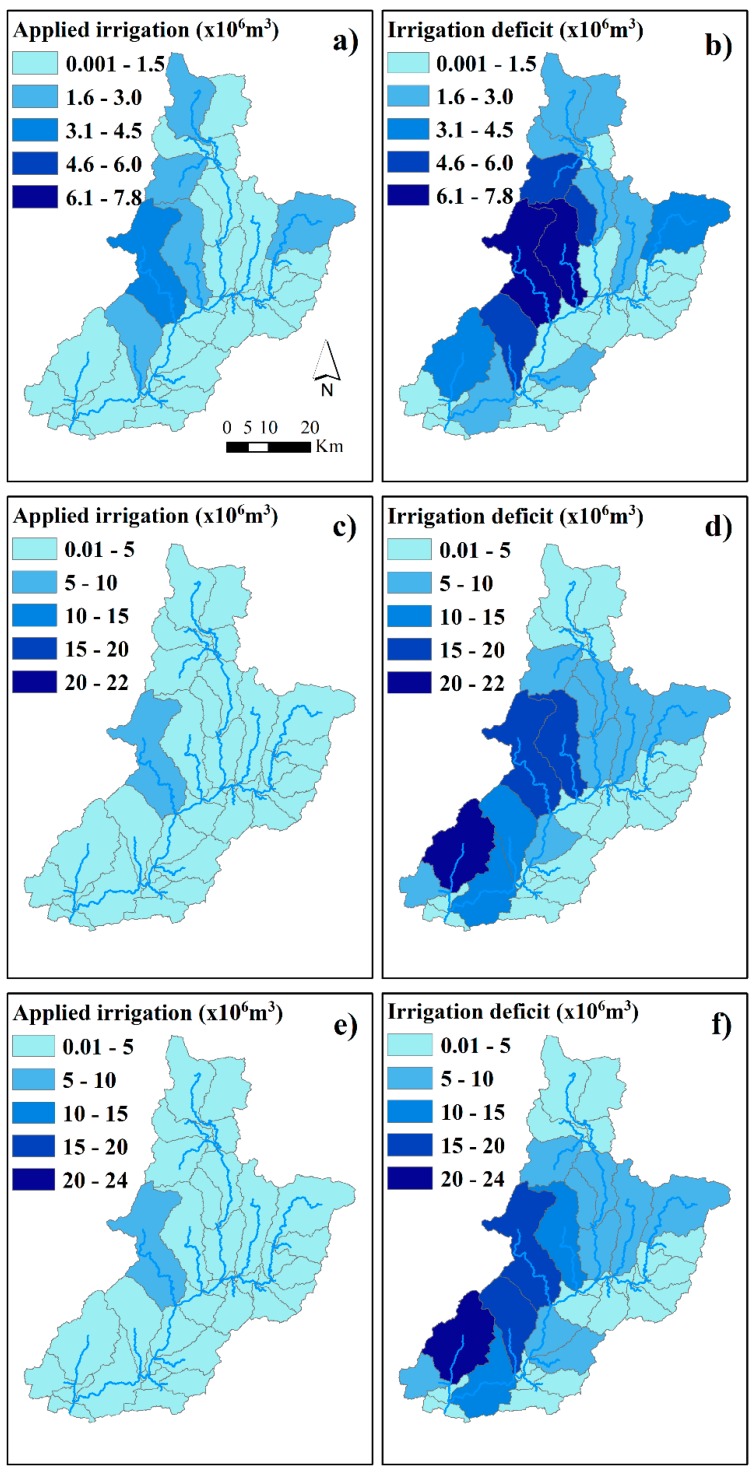
The applied irrigation and irrigation deficit at sub-basin scale in the Sabor River basin. (**a**) and (**b**) represent the applied irrigation and the irrigation deficit in the historical view, respectively. Figures (**c**) and (**d**) represent the data on the scenario of an increase in the irrigated area, and (**e**) and (**f**) represent the data on the scenario of a projection to 2060.

**Figure 9 ijerph-16-02419-f009:**
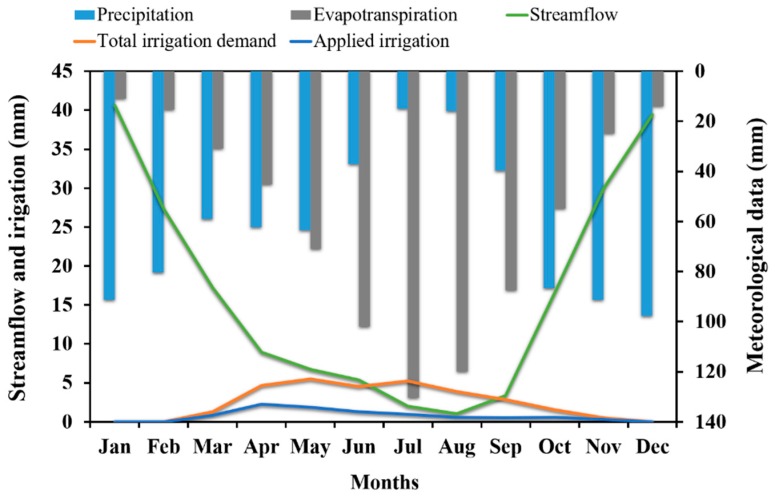
The streamflow, total irrigation demand, applied irrigation, precipitation, and evapotranspiration in the Sabor River basin for the period 1960–2008.

**Figure 10 ijerph-16-02419-f010:**
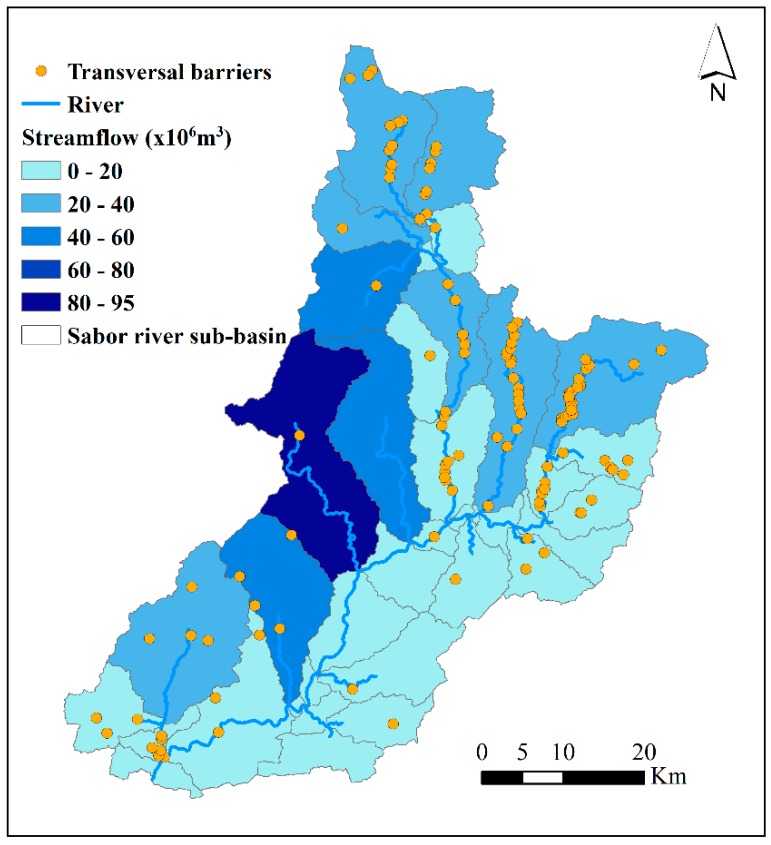
The spatial distribution of dams and weirs in the Sabor River basin and the annual average streamflow between 1960 and 2008.

**Table 1 ijerph-16-02419-t001:** The resident population in 1960 and 2011 and the variation in population per municipality according to Statistics Portugal [19]. The population density in 1960 and 2011 and the domestic water consumption per capita between 1995 and 2009 per municipality in the Sabor River basin. The domestic water consumption per capita is available at PORDATA [20]. The legend is: VP 1960–2011 (%), the variation in population between 1960 and 2011 (%); DWCPC 1995–2009 (L·day^−1^), the domestic water consumption per capita between 1995 and 2009 (L·day^−1^).

Municipality	Resident Population in 1960	Resident Population in 2011	VP 1960–2011 (%)	Population Density in 1960 (inhabitants·km^−2^)	Population Density in 2011 (inhabitants·km^−2^)	DWCPC 1995–2009 (L·day^−1^)
Alfândega da Fé	9659	5104	−47	30	16	43
Bragança	37,556	35,341	−6	32	30	44
Carrazeda de Ansiães	14,326	6373	−56	51	23	45
Freixo de Espada à Cinta	7252	3780	−48	30	15	55
Macedo de Cavaleiros	26,219	15,776	−40	38	23	47
Miranda do Douro	18,952	7482	−61	39	15	39
Mogadouro	19,626	9542	−51	26	13	74
Torre de Moncorvo	18,765	8572	−54	35	16	54
Vila Flor	11,829	6697	−43	45	25	43
Vimioso	12,763	4669	−63	27	10	55
**Total**	**176,947**	**103,336**	**−42**	**34**	**20**	**50**

**Table 2 ijerph-16-02419-t002:** The crops that were identified in the Sabor River Basin and their irrigable area and irrigated area according to agricultural census of 1999.

Crop	Irrigable Area (km^2^)	Irrigated Area (%)	Irrigated Area (km^2^)
Maize	28.5	90	25.7
Potato	28.5	80	22.8
Vineyards	43.4	7	3.04
Olive groves	321.8	7	22.5
Fruit trees	130.8	70	91.5
Forages	1.8	5	0.09
Horticulture	31.9	50	15.9
**Total**	**586.7**	**-**	**181.5**

**Table 3 ijerph-16-02419-t003:** The data input into SWAT and MIKE HYDRO Basin.

Data Type	Description	Source
Topography	Digital Elevation Model (10 m)	Directorate-General of Territory http://www.dgterritorio.pt/
Land use	Corine Land Cover 1990, 2000 and 2006 (1:100,000)	European Environment Agency http://www.eea.europa.eu/
Soil type	Soil map of Trás-os-Montes and Alto Douro (1:100,000)	Directorate-General of Territory http://scrif.igeo.pt/
Meteorology	Daily precipitation, maximum and minimum temperatures, solar radiation, relative humidity, and wind speed	National System of Water Resources Information https://snirh.apambiente.pt/
Hydrography	Daily streamflow between 1957 and 2008	National System of Water Resources Information https://snirh.apambiente.pt/
Resident population	Number of inhabitants per year and per municipality between 1960 and 2008	Statistics Portugal https://www.ine.pt/
Water consumption	Domestic water consumption per municipality (L·day^−1^·inabitant^−1^) for the years 1995, 2001, 2006, 2008 and 2009	Data base of Portugal https://www.pordata.pt/
COS2007	Land use and land cover of 2007	Directorate-General of Territory http://www.dgterritorio.pt/
COS2015	Land use and land cover of 2015	Directorate-General of Territory http://www.dgterritorio.pt/
FAO 56	FAO-56 Dual Crop Coefficient method. Tables of chapters 6, 7 and 8	Food and Agriculture Organization of the United Nations http://www.fao.org/home/en/
Projection of the resident population	Projection of number of inhabitants for 2060	Statistics of Portugal https://www.ine.pt/
Projection of the irrigable area of olive groves	Projection of the irrigable area of olive groves for 2060 based on increased between COS2007 and COS2015	Directorate-General of Territory http://www.dgterritorio.pt/

**Table 4 ijerph-16-02419-t004:** Crop parameters of the Sabor River basin that were input into MIKE HYDRO Basin and provided by the FAO tables [54]. The parameters include the crop stages, sowing date, basal crop coefficient (*K_cb_*), root depth, maximum height, and depletion fraction for each crop. Descriptions of the parameters can be found at http://www.fao.org/.

Parameters	Maize	Potato	Vineyards	Olive Groves	Fruit Trees	Forages	Horticulture
**Crop stages (days)**							
Initial	30	30	30	30	35	10	23
Development	40	35	60	90	60	20	33
Middle	50	50	40	60	110	144	39
Late	30	30	80	90	61	23	21
**Sowing date**	1-Apr	1-Apr	21-Feb	1-Mar	1-Mar	15-Oct	1-Apr
***K_cb_***							
*K_cb_* initial	0.15	0.15	0.15	0.55	0.47	0.3	0.15
*K_cb_* middle	1.15	1.1	0.65	0.65	0.76	0.7	1.02
*K_cb_* late	0.33	0.65	0.4	0.65	0.64	0.7	0.72
**Maximum height (m)**	2	0.6	1.75	4	3.67	0.31	0.45
**Root depth**							
Initial (m)	0.3	0.3	1	1	1	0.3	0.25
Middle (m)	1	0.4	1.5	1.45	1.45	0.65	0.56
**Depletion fraction (*p*)**	0.55	0.35	0.45	0.65	0.51	0.54	0.42

**Table 5 ijerph-16-02419-t005:** The goodness-of-fit indicators for the calibration, at a daily time-step for the period 1960–1999, and the validation, for the period 2000–2008, of streamflow in the Sabor River basin. Symbols: R^2^—coefficient of determination, NSE—Nash–Sutcliffe coefficient, RSR—the ratio of the root mean square error to the standard deviation of measured data, and PBIAS—the percent of bias.

Measure	Calibration	Acceptable Ranges	Validation	Acceptable Ranges
R^2^	0.63	>0.5 acceptable [38]	0.80	>0.75 very good [38]
RSR	0.62	Satisfactory [38]	0.63	Satisfactory [38]
NS	0.62	Satisfactory [38]	0.61	Satisfactory [38]
PBIAS	2.7%	Very good [38]	−24%	Satisfactory [38]

**Table 6 ijerph-16-02419-t006:** The irrigation demand in the Sabor River basin and the reference irrigation allocation for each crop. The simulation was performed at the sub-catchment scale and for the period 1960–2008. The reference irrigation allocation for each crop was obtained from the Directorate-General for Agriculture and Rural Development (DGARD) [55] and Statistics of Portugal (National Statistics Institute, INE) [17]. The DGARD provides a reference irrigation allocation for groundwater and for different surface irrigation methods. We used an average of the sprinkler, micro-sprinkler, and cannon irrigation methods for the maize, potato, forages, and horticulture crops, and the drip irrigation method was used for the vineyard, olive grove, and fruit tree crops. The reference irrigation allocation from (DGARD) is for the Northern and Central Interior Region of Portugal, and the reference irrigation allocation from Statistics Portugal is for Portugal.

Irrigated Crops	Sabor River Basin	DGARD			INE
	Irrigation Demand (m^3^∙ha^−1^)	Sprinkler, Micro-Sprinkler and Cannon (m^3^∙ha^−1^)	Drip (m^3^∙ha^−1^)	Groundwater (m^3^∙ha^−1^)	Irrigation (m^3^∙ha^−1^)
Maize	5308	9163		7546	6177
Potato	5335	5484		4712	7385
Vineyards	2436		2281	2147	2302
Olive groves	2783		2919	2748	2259
Fruit trees	5620		6482	6100	5817
Forages	3475	5115		4213	8823
Horticulture	4496	5404		4514	5574

**Table 7 ijerph-16-02419-t007:** The average annual quantity of used water and the water demand deficit (for domestic consumption) simulated in the Sabor River Basin for the historical view (1960–2008), and for the two scenarios (an increase in the irrigated area and the projection to 2060). The scenario of an increase in the irrigated area represents the irrigation of the total irrigable area identified in Continental Portugal for 2007 (COS2007) [21].

Historical View/Scenarios	Total Water Demand (10^3^ m^3^)	Used Water (10^3^ m^3^)	Water Demand Deficit	
			(10^3^ m^3^)	%
Historical view from 1960 to 2008	1372.27	950.47	421.8	31
The scenario of increase of irrigated area	1372.27	866.42	505.62	37
Scenario of projection to 2060	1044.5	659.33	385.15	37

**Table 8 ijerph-16-02419-t008:** The average annual irrigation demand, applied irrigation, and irrigation deficit per crop in the Sabor River Basin for the historical view (from 1960 to 2008) and for the two scenarios (an increase in the irrigated area and a projection to 2060). The scenario of an increase in the irrigated area represents the irrigation of the total irrigable area identified in COS2007 [21].

Irrigated Crops	Area (km^2^)	Irrigation Demand (10^6^ m^3^)	Applied Irrigation (10^6^ m^3^)	Irrigation Deficit	
				(10^6^ m^3^)	%
**Historical view from 1960 to 2008**					
Maize	25.7	14.08	4.14	9.94	67.2
Potato	22.8	12.77	3.7	9.06	67.8
Vineyards	3	0.75	0.3	0.45	56.6
Olive groves	22.5	6.18	2.34	3.84	57.2
Fruit trees	91.5	53.68	16.09	37.59	65.1
Forages	0.1	0.03	0.02	0.01	50.5
Horticulture	15.9	6.64	2.13	4.51	64.5
**Total**	**181.6**	**94.13**	**28.71**	**65.41**	**69.5**
**The scenario of increase of irrigated area**					
Maize	28.5	17.7	3.98	13.72	77.5
Potato	28.5	18.1	4	14.12	77.9
Vineyards	43	11	2.7	8.27	75.4
Olive groves	322	91.22	21.93	69.29	76
Fruit trees	131	81.33	17.78	63.55	78.1
Forages	1.8	0.64	0.28	0.35	55.4
Horticulture	32	17.26	4.05	13.22	76.6
**Total**	**586.8**	**237.2**	**54.71**	**182.5**	**76.9**
**The scenario of projection to 2060**					
Maize	25.7	15.38	3.74	11.63	77.6
Potato	22.8	13.97	3.34	10.63	78.1
Vineyards	3	0.77	0.20	0.57	69.8
Olive groves	494	140.08	33.10	106.98	70.6
Fruit trees	91.5	56.74	12.67	44.07	76
Forages	0.1	0.03	0.01	0.02	56.8
Horticulture	15.9	8.34	2.11	6.23	75.8
**Total**	**653**	**235.3**	**55.2**	**180**	**76.6**

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
