# Peer review of "Development of a Hydrologic and Water Allocation Model to Assess Water Availability in the Sabor River Basin (Portugal)"

_ijerph, 2019, doi:10.3390/ijerph16132419_

Round 1

Reviewer 1 Report

In general, there needs for major changes in the entire manuscript for its contents, details and edited by a native English editor in the entire manuscript. Please see the attached.

Author Response

Title

Title is too long and unclear, whether authors wanted to model water availability for supporting decision-making or to develop decision support system that improves decisions in the study basin.

I changed the title for “Development of a hydrologic and water allocation model to assess water availability in the Sabor River basin (Portugal)”

Abstract

Suggest to craft the abstract carefully because this did not encompass the key problem(s), key findings and recommendation.

I rewrote the abstract. I talked about the problem, conclusions and the importance of the study for Portuguese farmers.

This temporal period is for historical timeline, but did not explicitly tell a period for future scenarios. Is it 2010-2060?

I included the temporal period for future scenarios. From L23 to L25.

The temporal period is present in the Conceptual framework model sub-section from L171 to L179.  The explanation is: in the historical view, the simulation of the water used in domestic consumption and irrigation was between 1960 and 2008. The simulation for two scenarios was performed with the data of surface runoff, aquifer recharge, precipitation and reference evapotranspiration between 1960 and 2008. In the first scenario is simulated the water used in domestic consumption between 1960 and 2008 but in the irrigation was assumed that all irrigable area identified in the basin was irrigated. In the second scenario was simulated the expansion of the area of the olive grove and the decrease of the resident population for 2060.

what is the time step of data used?

The time step of simulations was on a daily basis between 1960 and 2008 in SWAT and MIKE HYDRO Basin. You can find it in Conceptual framework model sub-section, L178 and L179.

Introduction

- There needs for providing research gaps or current challenges or research problems in the study basin.

- Insufficient background information about the study basin.

- Missing a focus of the paper and justification for the study in the basin and how this study was important to be undertaken.

- Importantly need specific research question(s) relating to the topic.

I rewrote the introduction with the corrections, from L47 to L68.

This should be presented in the method section, not here.

Yes, the description of SWAT and MIKE Basin should be presented in the method section. But I deleted it because you told me I must simplify common information about the models (SWAT & MIKE HYDRO) in the method section.

No specific research questions and key problem(s) in the paper.

No specific focus on why this study is importance to the study area and at a certain context.

The answer was inserted in the introduction from L41 to L68

The problem in this basin is the lack of water in the dry season when the necessity of water for irrigation crops is higher. To solve this problem, the Portuguese government was created a National Irrigation Program. This way, it is essential to know if the water was produced in the basin is sufficient to satisfy the requirements of the crops, and how much water would be needed according to the increase in the irrigation area proposed by the National Irrigation Program.

Objectives

These three objectives can be summarized as sounded quite similar among them.

Suggest to reformulate the objectives of this study again as the current ones were addressed similarly.

I reformulate the objectives from L85 to L92

The sentence is the following:

“The specific aims of this work are: (i) to know if the water produced in the basin between 1960 and 2008 was sufficient to satisfy the requirements of the crops, as well as, the domestic consumption, (ii) to determine the quantity of water will be necessary to irrigate all irrigable area that exists in the basin (first scenario), and the decrease of 24% of the resident population and the expansion of 29% of olive groves area for 2060 (second scenario).”

Material and methods

It's better to start with a topo map then elevation map, pop density and agricultural land use map.

Coordinates are missing in these maps while labels are blurred.

I did. I start with a topo map then elevation map, population density and changed the irrigable crops by agricultural land use map. I only put the coordinates on the first map because the study area is always the same, and the coordinates also. But if you think is wrong I will put the coordinates in all figures of the document.

Already presented in the map so just highlight a few major crop areas are enough. Another way, author may consider to remove the map 3 and populate these crop areas into a table. This way the authors can avoid repeated information and use space for other necessary background information.

I did. In figure 1c, I changed the irrigable crops to agricultural land use map, and I inserted the crop areas into table 2. I inserted also the irrigated crops, and I explained the difference between the irrigable and irrigated area. I also represented the respective areas according to agricultural census of 1999.

Table 2. The crops identified in the Sabor River Basin and the respectively irrigable area and irrigated area according to agricultural census of 1999.

Crop

Irrigable   area (km2)

Irrigated   area (%)

Irrigated   area (km2)

Maize

28.5

90

25.7

Potato

28.5

80

22.8

Vineyards

43.4

7

3.04

Olive groves

321.8

7

22.5

Fruit trees

130.8

70

91.5

Forages

1.8

5

0.09

Horticulture

31.9

50

15.9

Total

586.7

181.5

what is a temporal period for the baseline and future scenario?

The temporal period is present in the Conceptual framework model sub-section from L171 to L179.  The explanation is: in the historical view, the simulation of the water used in domestic consumption and irrigation was between 1960 and 2008. The simulation for two scenarios was performed with the data of surface runoff, aquifer recharge, precipitation and reference evapotranspiration between 1960 and 2008. In the first scenario is simulated the water used in domestic consumption between 1960 and 2008 but in the irrigation was assumed that all irrigable area identified in the basin was irrigated. In the second scenario was simulated the expansion of the area of the olive grove and the decrease of the resident population for 2060.

This is confusing when authors said earlier that 1960-2009 was the historical period of the study but a range of data covered to 2011. What was the actual temporal period of the baseline?

The temporal period of the baseline is 1960-2008. In table 1, I put the population density of 2011, because the census of population was realized in 2011. This information was only used to determine the decrease of population density in the basin.

The temporal period of the baseline is 1960-2008, not 1960-2009. It was my mistake, I put the wrong date.

Conceptual framework model section

Need a clearer outline for this section. May consider to use sequential order by starting off with data used, model set up, scenario development with description of baseline and future, and model calibration.

I rewrote all section with the corrections, from L159 to L179.

First mention this, there needs for a full name.

HRU means hydrologic response units from SWAT. This word was deleted because you told me I must simplify common information about the models (SWAT & MIKE HYDRO) in the method section.

Grammatic errors as mixed present tense with past tense. See these errors throughout the paper so need a native English speaker to help editing the paper.

The English will be reviewed by MDPI editing services.

Such common information about the models (SWAR & MIKE HYDRO) can be simplified so authors can then provide a brief description regarding key features of the selected models.

I did. I review the methods and I deleted some information in “SWAT hydrological model and calibration” and “MIKE HYDRO Basin model” to simplify this section.

Missing coordinates while resolution of the map needs for a higher value.

I only put the coordinates on the first map because the study area is always the same, and the coordinates also. But if you think is wrong I will put the coordinates in all figures of the document. The resolution of the maps in this document is weak, but I export the maps from to ArcMap with a good resolution (in 600 dpi). I think you can find it in the attachment.

Seems like authors meant water users' needs in the study area, while mixing water demand (consumption plus leakage at supply side) with this section. Better make sure about this.

Yes. I corrected it from L230 to L232. The sentence is the following:

“In the regular water user was inserted the amount of water used in domestic consumption, which was calculated with resident population and water consumption per municipality. The resident population is available per year between 1960 and 2008 at INE [19].”

May consider to populate all these data into a table and describe what are assumptions and key elements between the baseline and future scenarios. Missing background information on biophysical variables that were used in exploring future scenarios such as precipitation and temperature.

I did. The answer was inserted in the Historical view and evolutionary scenarios sub-section from L315 to L332.

“In the historical view was modelled the domestic consumption and irrigation between 1960 and 2008. In this simulation was used the percentage of irrigated area per crop showing in Table 2 The values of irrigated area per sub-basin are found in supplementary material, worksheets 3. The areas were inserted in each of the sub-basin of the irrigated module of MIKE HYDRO Basin.

In the first scenario was modelled the domestic consumption, and a time series of surface runoff, aquifer recharge, precipitation and reference evapotranspiration between 1960 and 2008, and in irrigation was assumed that all irrigable area identified in COS 2007 was irrigated (Table 2). The area of each irrigable crops per sub-basin are found in supplementary material, worksheets 4.

In the second scenario was modelled both demographic and irrigation projection for 2060 with a time series of surface runoff, aquifer recharge, precipitation and reference evapotranspiration between 1960 and 2008. The demographic projection was based in the technical report of the resident population between 2012 – 2060 to the North zone and to the central projection scenario [43]. The projection estimates a decrease of 24% of the resident population until 2060. Figure 4a shows the gradual reduction of the resident population between 2012 – 2060 for all municipalities into the Sabor River basin. The projection estimates a decrease of 25056 inhabitants in municipalities of the basin for 2060, which means less 5 inhabitant.km2. The municipalities with the largest population reduction will occur in Bragança and Macedo de Cavaleiros with 8495 and 3793 inhabitants respectively. In the other municipalities, the reduction of population ranged between 910 and 2296 inhabitants.”

No reference for national program and demographic projection.

I did. The answer was inserted in the Historical view and evolutionary scenarios sub-section in L325 and L326.

The demographic projection was based in the technical report of the resident population between 2012 – 2060 to the North zone and to the central projection scenario [40].”

Temporal period for baseline is not consistent with the period in the abstract and other parts of the paper.

I corrected it. In abstract, Conceptual framework model and Historical view and evolutionary scenarios sub-sections.

Results

better explicitly indicate a temporal period for future scenario.

what is the temporal period for this scenario?

I corrected it. In Figure 7, L427 and L428 and in Figure 8, L460 and L461.

“The scenario of increase of irrigated area represents the irrigation of all irrigable area identified in COS 2007 [21].”

Reviewer 2 Report

The paper presents current water scarcity problems resulting from temporal and spatial variability of water resources and demand. The authors pointed out the future problems that will have to be faced in the context of satisfying water needs (irrigation and domestic consumption). The Authors presented a decision support system (the interaction between SWAT and MIKE HYDRO Basin) that allows effective water management. The paper is very well written. The results have been clearly presented, described in detail and discussed against the background of results obtained by other researchers.

Comments:

·         Table 3. Change Abril to April

·         Table 3 The descriptions in the column Sowing date should be unified: 1 Apr, 1 Apr, 21 Feb, 1 Mar, 1 Mar, 15 Oct, 1 Apr,

·         Page 14, line 11: change 1960 c 1999 to 1960 – 1999,

·         It is a pity that the Authors do not have data from meteorological stations located outside the catchment area in east and south-east directions. Could it have affected the results of precipitation interpolation?

·         Page 16, line 28: The abbreviation INE must be completed in the description of the table „...Agriculture and Rural Development (DGARD) [39] and Statistics Portugal (INE) [29]”,

·         Authors should complete the method of precipitation interpolation in the SWAT model.

Author Response

Review 2

· Table 3. Change Abril to April

· Table 3 The descriptions in the column Sowing date should be unified: 1 Apr, 1 Apr, 21 Feb, 1 Mar, 1 Mar, 15 Oct, 1 Apr,

· Page 14, line 11: change 1960 c 1999 to 1960 – 1999,

I did the corrections. Now this table is number 4 in L262.

· It is a pity that the Authors do not have data from meteorological stations located outside the catchment area in east and south-east directions. Could it have affected the results of precipitation interpolation?

Yes, It is pity do not have data from meteorological stations located outside the catchment area. I believe it influences the results, but not significantly because surrounding the catchment area the weather conditions did not change a lot.

· Page 16, line 28: The abbreviation INE must be completed in the description of the table „...Agriculture and Rural Development (DGARD) [39] and Statistics Portugal (INE) [29]”,

Yes, I did the corrections from L395 and L397. The sentence is the following:

The reference irrigation allocation for each crop by Directorate-General for Agriculture and Rural Development (DGARD) [39], and Statistics of Portugal (National Statistics Institute - INE) [17].

· Authors should complete the method of precipitation interpolation in the SWAT model.

Yes, I inserted in L201 and L202 the following sentence:

The precipitation data of eleven station and the meteorological data of Folgares station (06N/01C) (Figure 1c), which are available at the National System of Water Resources Information [11] were inserted in the SWAT model. The Thiessen polygon method was selected to interpolate precipitation into the basin.”